# Calcium-Permeable Channels Cooperation for Rheumatoid Arthritis: Therapeutic Opportunities

**DOI:** 10.3390/biom12101383

**Published:** 2022-09-27

**Authors:** Hong-Yu Liang, Huan-Xin Yin, Shu-Fang Li, Yong Chen, Ying-Jie Zhao, Wei Hu, Ren-Peng Zhou

**Affiliations:** 1The Second School of Clinical Medicine, Anhui Medical University, Hefei 230032, China; 2Department of Clinical Pharmacology, The Second Hospital of Anhui Medical University, Hefei 230601, China; 3The Key Laboratory of Anti-Inflammatory and Immune Medicine, Anhui Medical University, Hefei 230032, China

**Keywords:** rheumatoid arthritis, calcium-permeable channel, acid-sensitive ion channel, transient receptor potential channel, P2X receptor

## Abstract

Rheumatoid arthritis is a common autoimmune disease that results from the deposition of antibodies–autoantigens in the joints, leading to long-lasting inflammation. The main features of RA include cartilage damage, synovial invasion and flare-ups of intra-articular inflammation, and these pathological processes significantly reduce patients’ quality of life. To date, there is still no drug target that can act in rheumatoid arthritis. Therefore, the search for novel drug targets has become urgent. Due to their unique physicochemical properties, calcium ions play an important role in all cellular activities and the body has evolved a rigorous calcium signaling system. Calcium-permeable channels, as the main operators of calcium signaling, are widely distributed in cell membranes, endoplasmic reticulum membranes and mitochondrial membranes, and mediate the efflux and entry of Ca^2+^. Over the last century, more and more calcium-permeable channels have been identified in human cells, and the role of this large family of calcium-permeable channels in rheumatoid arthritis has gradually become clear. In this review, we briefly introduce the major calcium-permeable channels involved in the pathogenesis of RA (e.g., acid-sensitive ion channel (ASIC), transient receptor potential (TRP) channel and P2X receptor) and explain the specific roles and mechanisms of these calcium-permeable channels in the pathogenesis of RA, providing more comprehensive ideas and targets for the treatment of RA.

## 1. Introduction

Rheumatoid arthritis (RA) is an autoimmune disease of unknown cause that affects several organs and tissues, including synovial joints and can involve multiple systems throughout the body. RA affects approximately 1% of the population and has a male-to-female ratio of 2.5:1 [1]. From the pathological perspective, the pathogenesis of RA mainly includes three aspects: synovitis, structure destruction and inflammation infiltration. Synovial and bone changes have been reported in some patients as early as four months after the onset of the disease. In terms of clinical features, patients often experience joint deformity, morning stiffness and pain, swelling and hyperplasia of the synovial membrane, increased C-reactive protein in the blood and weakness due to impaired motor function [1,2,3,4,5]. In addition to affecting the locomotor system, novel clinical signs of RA have also been identified, such as skin and respiratory involvement [6,7]. In terms of treatment, there has been considerable progress in the drug treatment of RA. The classical RA treatment drugs include non-steroidal anti-inflammatory drugs (NSAIDs) such as aspirin, glucocorticoids such as dexamethasone, and disease-modifying antirheumatic drugs (DMARDs) such as methotrexate. These drugs can relieve the condition and the pain of RA, but, due to the multiple adverse reactions to NSAIDS [8,9] and adverse effects of DMARDs that have been clinically reported [10,11], it has become more to explore new therapeutic targets for RA.

Calcium ions (Ca^2+^) affect almost every aspect of cellular life. As an important second messenger for cellular signaling, there is a huge difference in concentration between intracellular (~100 nM free) and extracellular (mM) Ca^2+^ levels, and the cell devotes a large amount of energy to maintaining this difference in concentration [12]. Clearly, the large difference in intracellular and extracellular Ca^2+^ concentration provides the basis for rapid changes in [Ca^2+^]_i_, which conduct the intracellular signal transduction [12,13]. Ca^2+^ is a central aspect of inflammation as well as inflammatory diseases. In fact, Ca^2+^ is thought to be closely associated with the release and metabolism of inflammatory substances such as histamine, prostaglandins and leukotrienes by mast cells during the outbreak of inflammation [14]. It has been suggested that Ca^2+^ can regulate the metabolism of arachidonic acid in T cells and mediate synovial inflammation in RA patients during the inflammatory response [15]. In addition, there is a large body of data showing that Ca^2+^ mediates the infiltration of large numbers of immune cells during the development of RA, leading to an uncontrolled inflammatory response [16]. These results indicate that Ca^2+^ itself plays a core role in the inflammatory response, particularly in RA. Ca^2+^ channels, as a main driver of calcium signaling transduction that are widely expressed on the surface of cell membranes, endoplasmic reticulum membranes and mitochondrial membranes, mediate the influx of extracellular Ca^2+^ or the release of Ca^2+^ from Ca^2+^ stores. Based on the gating mechanism of these channels, Ca^2+^ channels can be classified as: (1) voltage-gated Ca^2+^ channels, (2) ligand-gated channels, (3) store-operated channels (SOC), (4) second messenger-operated channels, (5) acid-sensing ion channels, and (6) mechano-gated channels [17].

Ca^2+^ channels, therefore, are particularly important in all life activities. Indeed, Ca^2+^ channels have long been a therapeutic target for various diseases. It is well-established that the moderate inhibition of Ca^2+^ channels in cardiac and vascular smooth muscle cells can inhibit myocardial contraction and vasodilation, thereby relieving tachycardia and angina pectoris [18,19,20]. In respiratory diseases such as asthma, the blockade of Ca^2+^ channels can inhibit bronchial ground constriction [17,21]. In addition, there has been increasing evidence in recent years that calcium-permeable channels play an important role in joint diseases such as RA. In RA, the modulation of different families of Ca^2+^ channels has been shown to regulate the progression of RA. For example, our previous study found that the inhibition of acid-sensitive ion channels (a calcium-permeable channel) can slow arthritic rat chondrocyte apoptosis [22,23]. Here, we briefly describe the pathogenesis of RA, review the Ca^2+^-permeable channels that play an important role in RA, and detail the mechanisms of Ca^2+^-permeable channels in the pathogenesis of RA, providing novel therapeutic targets for RA (Figure 1).

## 2. Pathology of Rheumatoid Arthritis (RA)

RA is defined as a systemic autoimmune disease associated with a chronic inflammatory process. RA can damage joints and extra-articular organs, including the heart, kidneys, lungs, digestive system, eyes, skin, and nervous system. The main joint manifestations can be divided into three areas, including synovitis, structural damage to joints, and the activation of the inflammatory response [24].

Inflammation of the synovium due to reflex immune activation in RA joint-swelling is characterized by the infiltration of leukocytes into the normally sparse synovium [25]. The aggregation of leukocytes is not due to proliferation but is the result of the chemotaxis of circulating leukocytes. There are many possible reasons for the leukocyte infiltration that occurs in the synovium in RA, including: (1) adaptive immune pathways, (2) activation of the innate immune system, (3) cytokines and intracellular signaling pathways, and (4) mesenchymal tissue response [26,27,28]. Within a short time after onset, the synovium can thicken to 10 cells deep and is mainly composed of type A (macrophage-like) and type B (fibroblast-like) synovial cells (FLSs) [1]. The infiltrating immune cells produce a variety of cytokines that activate FLSs. Under the action of various mediators, activated FLSs apoptosis is reduced, while it can proliferate in large numbers. The exact mechanism regarding synovial cell proliferation may be related to the induction of certain growth factors such as TGF-β or to oncogenes [29,30]. Recently, the process of interaction between synovial macrophages and FLS has also been revealed, which may suggest the existence of a mechanism for the development of chronic inflammation in RA [31], but the exact mechanism still needs to be further explored. In this way, the inflammation of the synovial membrane, as well as the proliferation of the landscape, is well-described.

The erosion of cartilage and bone is the main sign of structural damage in RA, and the quantification of bone and cartilage damage has been an important tool in the clinical diagnosis of the disease and to monitor the efficacy of drug therapy [32]. However, two different mechanisms are involved in the erosion of bone and cartilage [30,32]. Bone erosion tends to occur in the part of the synovial membrane that is in contact with the bone, i.e., the bone around the joint is more susceptible to erosion [32,33]. Indeed, after the onset of RA, synovial inflammation can lead to local hypoxic conditions and cytokine-induced neovascularization, increasing the microvasculature in the interstitial space [34]. The microvasculature located in these interstitial spaces contributes to the homing of osteoclasts. Osteoclasts consist of multinucleated giant cells containing from 2 to 50 closely packed nuclei, mainly on the surface of the bone, around the vascular channels in the bone. Osteoclasts mainly perform bone resorption functions and functionally correspond to osteoblasts. The high expression of anti-tartaric acid phosphatase and histone K are the main markers of osteoclasts. Current research on osteoclast activation has focused on the regulation of the RANK/RANKL/OPG axis. Osteoclast formation is regulated by a variety of factors, and NFATc1, which was first identified in T cells, is one of the most critical and widely studied transcription factors [35]. When osteoclasts come into contact with bone and receive the relevant molecular signals, they are activated and osteoclast-mediated bone resorption leads to the loss of cortical bone [36]. The high-resolution CT of joints in RA patients also demonstrates the bone resorption role played by osteoclasts [37,38,39,40,41]. Articular cartilage consists of a non-mineralized superficial layer and a deep mineralized layer close to the bone. The disintegration of undifferentiated cartilage includes two aspects. One is that chondrocytes can secrete various mediators such as ADAMTS (a metalloproteinase with a thrombin reactive protein motif) and matrix metalloproteinase (MMP) after inflammatory stimulation, leading to the degradation of the perichondral matrix. Secondly, FLSs and infiltrating immune cells can also secrete MMPs and other inflammatory cytokines to degrade the surrounding cartilage [30]. In addition, the synovium invades the periarticular bone at the cartilage–bone junction due to a persistent chronic inflammatory state, leading to bone erosion and cartilage degeneration. Recently, a number of in vivo studies have demonstrated that the modulation of FLSs can alleviate cartilage damage in RA and suggested a number of potential drug targets [42,43].

A sustained inflammatory response occurs when autoantigen–antibody complexes are deposited in the joint and become difficult to remove. The onset of inflammation is often the main driver of the clinical manifestations of RA. Localized inflammation in the joints often leads to redness, swelling and pain, and a high level of inflammatory cytokines can be detected in the joint cavity. At this point, large numbers of immune cells such as dendritic cells, neutrophils, lymphocytes, T cells, and mast cells infiltrate the joint and release large amounts of inflammatory factors. Dendritic cells often play the role of “sentinel” when inflammation occurs, which can activate the immune system, processing and presenting self-antigens to naive T cells. During the development of RA, the migration of dendritic cells towards the site of inflammation is mediated by the chemokine CCR6 and increased CCR6 expression on the surface of DCs during RA pathogenesis, whose ligand CCL20 is highly expressed in synovial tissue. Once attracted to the joint, mature DCs produce cytokines IL-12 and IL-23, which promote antigen-specific Th17 responses, leading to an imbalance between Th1, Th2, and Th17 responses and the release of inflammatory factors [24,44]. This initiates the outbreak of inflammatory cytokines at the joint. In fact, synovial proliferation and cartilage erosion are both based on inflammatory eruptions, and the release of inflammatory factors can promote synovial tissue proliferation and cartilage destruction; the three are not completely independent, but are linked.

## 3. Acid-Sensitive Ion Channels (ASIC)

Why can acid induce pain? Why can neurons sense changes in extracellular pH? These questions drove the search for acid sensors in the body. The first proton receptors located on nerve cell membranes were discovered in 1980s [45]. Subsequently, R. Waldmann et al. cloned one of these proton receptors for the first time and named it an acid-sensitive ion channel (ASIC) [46]. ASICs are degenerin/epithelial sodium channel superfamily and, to date, seven subtypes of ASIC channels have been identified. These include ASIC1a, ASIC1b, ASIC1b2, ASIC2a, ASIC2b, ASIC3, and ASIC4. When activated under acidic conditions, they all mediate the inward flow of a variety of cations, including Ca^2+^.

### 3.1. ASIC1a

As an acid-sensitive channel, ASIC1a is mainly located in the axon terminals of the nervous system. In recent years, it has also been found in tumors, bone tissues and other non-nervous systems. ASIC can be activated by a variety of acidifying pathological environments, including inflammation, tumor and ischemia and other processes, and ASIC1a plays an important role in a variety of pathophysiological processes. In the inflammatory response, local acidification due to the accumulation of inflammatory metabolites in the microenvironment leads to the activation of ASIC channels. In addition, in ischemic diseases such as ischemic stroke, ASIC1a can also mediate acid toxicity, leading to nerve damage [47].

The typical RA lesion is synovial inflammation, which can strike in the early stages of RA. Synovitis often leads to hyperplasia and tissue acidification of the synovial membrane. This provides a pathological basis for the role of the ASIC1a channel in RA synovitis. Indeed, it has been shown that the opening of ASIC1a channels activates Ras-related C3 botulinum toxin substrate 1 and promotes RA synovial cell invasion [48]. In addition, recent evidence has also demonstrated that ASIC1a promotes the proliferation of RA FLS cells through the ERK/MAPK and NFAT signaling [49,50]. This available evidence suggests that ASIC1a plays a toxic role in synovial inflammation, and the proper blockade of ASIC1a channels in the synovium may slow the progression of RA synovitis.

As described in the previous section, cartilage damage during RA is mainly attributed to the apoptosis of chondrocytes and degradation of the extracellular matrix. Ca^2+^, due to their unique physicochemical properties, are one of the most important signals in apoptosis and, therefore, calcium-permeable channels, which are responsible for their transport, are an important target when controlling cartilage damage. In the various pathological processes of RA, inflammatory bursts in the tissues lead to a decrease in pH within the local cartilage tissue [51]. ASIC1a has been shown to play an important role as an acid sensor in RA cartilage injury. Chen et al. first proposed that the inhibition of ASIC channels using amiloride reduced cartilage damage as well as chondrocyte apoptosis in rats with adjuvant arthritis (AA, an animal model of RA), identifying the chondroprotective effects of ASIC channel inhibitors [52]. However, due to the low selectivity of amiloride, it was difficult to identify the specific subtype of ASIC, although it is certain that this reveals a direct role of ASIC channels in chondrocyte apoptosis and creates a path for subsequent studies. In vitro studies have demonstrated that when ASIC1a is activated by extracellular acidification, chondrocyte extracellular matrix expression is reduced, and it is proposed that this may be due to a strong increase in [Ca^2+^]_i_ activation downstream of MAPK/ERK signaling as a result of ASIC1a activation [53]. This reveals the mechanism through which ASIC1a affects chondrocytes and provides new insights into cartilage damage in RA. Since then, a number of studies have identified a role for the ASIC1a channel in promoting apoptosis in chondrocytes as well as FLSs, possibly through the mitochondrial pathway [54], and the phosphorylation of extracellular-signal-regulated kinases [55]. In addition to apoptosis, pyroptosis is also a form of programmed cell death, characterized by the activation of caspase-1 and the secretion of the pro-inflammatory cytokine IL-1β/18. Our recent study reports that the ASIC1a-mediated elevation of [Ca^2+^]_i_ may lead to the pyroptosis of rat chondrocytes through the activation of calpain-2/calcineurin [56,57]. These studies have enriched the mechanisms by which the ASIC1a channel mediates cartilage damage, but relevant in vivo experiments are still lacking. However, our previous studies have shown that IL-6 significantly upregulates ASIC1a levels in chondrocytes of rats with adjuvant arthritis in a dose-dependent manner and significantly mediates chondrocyte apoptosis via ASIC1a [58]. In addition to IL-6, IL-1β and TNF-α also upregulated chondrocyte apoptosis in an NF-κB-dependent manner through the upregulation of ASIC1a [23]. These studies reveal the role that joint inflammation plays in cartilage damage, which is bridged by ASIC1a. In any case, the inhibition of ASIC1a channels at the pharmacological level using reasonable inhibitors has the potential to reduce chondrocyte apoptosis or scorch death. This provides new targets and options for RA treatment.

In adults, bone tissue is balanced by the continuous resorption of osteoclasts and the production of osteoblasts. The enhanced osteogenesis has a mitigating effect on the RA bone damage. Osteoclasts originate from monocytes and are found in the blood and circulation. The process of differentiation for osteoclasts is mainly regulated by the macrophage colony-stimulating factor and NF-κB receptor activator ligand (RANKL) [59]. Studies have shown that extracellular acidosis increases the activation and formation of osteoclast precursor cells and multinucleated osteoclast-like cells [60]. As a pH receptor, ASIC plays an important role in osteoclast activation induced by extracellular acidification. It has been reported that acidosis activates ASIC1a in osteoclasts as well as osteoclast precursor cells and promotes osteoclast migration and adhesion via integrin/Pyk2/Src signaling [61]. Cell activation and migration are important aspects of osteoclast-mediated bone resorption, and this study suggests an important role for ASIC1a in osteoclast-mediated bone destruction. However, the limitation is that the study was only performed under a low pH condition (pH = 6.0) and it was difficult to determine the effect of the pH gradient on osteoclast migration. In addition, there is a lack of data on the effect of low pH on the migration and adhesion of osteoclast precursor cells in vivo. Elevated [Ca^2+^]_i_ can also regulate osteoclast differentiation through a variety of pathways, such as nuclear factor of activated T cells c1 (NFATc1), etc. [62]. The elevated [Ca^2+^]_i_ mediated by ASIC1a activation is well known. Indeed, Li et al. showed that ASIC1a activation mediates osteoclast genesis via the NFATc1 pathway and that can be inhibited by the ASIC1a antagonist PcTX1 [63]. These studies point to an important role for ASIC1a in acid-induced osteoclast genesis, activation and migration, providing an important therapeutic target in osteoclast activation-related diseases such as RA and osteoporosis.

Local inflammation during RA is often characterized by swelling of the joint, pain and an outbreak of inflammatory factors in the synovial cavity. It has been shown that in RA synovial tissue, activation of ASIC1a mediates an increase in [Ca^2+^]_i_ and promotes the release of more inflammatory cytokines such as MIP-1a and IL-8 from FLSs via the NFATc3/RANTES pathway [49]. Although the role of ASIC1a in the release of inflammatory factors in RA is well established, the mechanism remains unclear and the evidence is still insufficient.

In conclusion, the toxic role of ASIC1a during RA is well established, i.e., overexpressed or activated ASIC1a promotes inflammation, structural damage and the release of inflammatory factors in RA synovium during the development of RA. Appropriate blockade of ASIC1a at the pharmacological level may be a novel strategy for the targeted treatment of RA, but the clinical trials that currently scarce. Therefore, further clinical evidence of the therapeutic significance of ASIC1a in RA and the need to develop more specific blocking agents are needed in the future.

### 3.2. ASIC2a

ASIC2a is mainly localized in the central nervous system and peripheral nervous system, and its pH_50_ is 4.5–4.9, similar to ASIC1a, and it also plays a certain role in the inflammatory response and the acid toxicity of the nervous system. Although the role of ASIC1a and ASIC3 in cartilage damage during RA is well established, interestingly, we found that overexpression of ASIC2a enhanced the chondroprotective effects of PcTx1 (an ASIC1a-specific inhibitor) and APETx2 (an ASIC3-specific inhibitor) and inhibited the ERK1/2 MAPK signaling pathway [64]. This is distinct from the other ASIC isoforms (ASIC1a/3) in cartilage injury and may be due to some unique physicochemical properties of ASIC2a, but in any case, this is the first suggestion of a role for ASIC2a in cartilage injury.

### 3.3. ASIC3

ASIC3 was mainly localized in the peripheral nervous system but was also detected in the trigeminal ganglion in the central nervous system. ASIC plays an important role in pain transmission due to its unique acid-sensitive current and its high expression in the nervous system. In fact, ASIC3-mediated currents were first detected in cardiac afferents and demonstrated that ASIC3 might be involved in pain transmission [65]. This also opened the research on ASIC3. At the physiological level, compared with other ASIC channels, ASIC3 can be opened at a pH closer to physiological conditions (pH_50_ = 6.4–6.6), and the transient inward current triggered by protons lasts longer [66]. This suggests that ASIC3 may play an important role in a variety of long-term chronic diseases, such as RA.

The main cause of synovial invasion in RA is the proliferation of intra-articular synovial cells, especially FLSs. Therefore, induction of apoptosis or inhibition of synovial cell proliferation can be a means to alleviate synovial invasion in RA. Hyaluronic acid strengthens the stability of synovial cells and prevents the spread of inflammatory mediators, thus reducing synovial inflammation. Sluka et al. identified ASIC3 channels in mouse knee synovial cells for the first time during their study of the role that ASIC3 plays in nociceptive sensitization in mouse joint inflammation [67], and their additional work revealed a role for ASIC3 in regulating the release of hyaluronic acid from FLSs [68]. This suggests an important role for ASIC channels in synovial tissue. In fact, the different subunits of the ASIC family play inconsistent roles in RA synovial invasion. It has been suggested that ASIC3 activation mediates a rise in [Ca^2+^]_i_ and induces apoptosis in the FLSs of RA patients [55]. Furthermore, in the collagen-induced arthritis model (an animal model that mimics RA in vivo), ASIC3 mediates the apoptosis of FLSs in response to intra-articular inflammatory mediators (e.g., IL-1β), and ASIC3^−/−^ mice exhibit marked synovial inflammation and invasion [69]. These reports suggest that ASIC3 channels may alleviate the synovial invasive process of RA by mediating calcium toxicity or inflammation, and thus inducing apoptosis in RA synovial cells. This differs from the previous toxic effects of ASIC1a in synovial inflammation, and there are many reasons for this contradiction, the most important of which may be the differences between ASIC isoforms that lead to this effect, the different cell types used, etc. To better understand the therapeutic opportunities that the ASIC3 channel provide in RA, a number of studies will be needed in the future to validate the impact of different modalities of modulation on RA.

In cartilage damage, it has been suggested for a decade that ASIC3 is expressed in FLSs as well as the chondrocytes and that it activates FLSs to release hyaluronic acid when extracellular pH is reduced (~5.5) [68]. This is the first indication of a role for ASIC family channels in the regulation of the extracellular matrix of articular cartilage, although the exact mechanism has not been revealed.

Indeed, early studies have shown that ASIC3 channels are activated during joint inflammation and that their activation is involved in the development of inflammatory pain [67,70,71,72]. Furthermore, in carrageenan-induced arthritis in mice (an acute arthritis model), ASIC3 expression in DRG neurons in joint nerve fibers was significantly upregulated (an approximately 50% increase) [70]. These results all suggest that a decrease in pH during joint inflammation may activate the ASIC3 channels expressed on innervated neurons, thereby transmitting pain signals to the center, manifesting as nociceptive hyperalgesia. In 2012, it was first suggested that, in mice with collagen-induced arthritis, ASIC3 knockout led to significant synovial inflammation and increased FLSs release of IL-6 and MMPs; however, interestingly, the mice showed less painful behavior [69]. This phenomenon is explained by the fact that ASIC3, which is localized at the nerve endings, plays an important role in pain transmission in RA, whereas ASIC3, which is located in synovial cells, plays a certain anti-inflammatory role. Therefore, when ASIC3 is knoced out, nociceptive transmission is reduced and inflammation in the synovial membrane is increased. This study led to an investigation of the role of ASIC channels in RA joint inflammation and also enriched the mechanism by which ASIC3 induces apoptosis in articular chondrocytes, possibly through the upregulation of FLSs to release MMPs. There may be some inhibitory effects of ASIC3 activation or overexpression on synovial invasion as well as an inflammatory burst in RA, but the role of other aspects of RA pathogenesis needs to be further explored. Due to its role in pain transmission and inflammation control, the potential significance of ASIC3 as a therapeutic target for RA is highlighted. Different modulations may have different effects, and more targeted drugs need to be developed in the future to precisely control ASIC3 channels during RA (Figure 2).

## 4. Transient Receptor Potential (TRP) Channels

In the last century, it was discovered that the deletion of a particular gene in fruit flies causes the onset of temporary visual impairment, so the researchers named this particular gene “transient receptor potential” (TRP) [73]. In 1993, TRP was recognized as a new superfamily of ion channels and its role in Ca^2+^ permeability was identified [74]. The first case of a human TRP homolog (TRPC1) was reported in 1995 [75]. To date, seven TRP families have been identified: TRPC, TRPV, TRPM, TRPA, TRPP, TRPML, and TRPN. An eighth TRP family was recently identified in *yeast* and named TRPY (Y for *yeast*) [76]. In this review, we focus on the TRPM, C and V subfamilies. It is worth mentioning that a fragment with α-kinase activity was fused to the C-terminus of TRPM6 and TRPM7, which is rare in other ion channel families. The research on its α-kinase function is limited.

### 4.1. TRPM7 Channel

The TRPM channel mainly consists of eight members, TRPM1-8. Interestingly, TRPM6 and TRPM7 form an additional serine/threonine-rich kinase domain at the C-terminus and an α-kinase domain that is homologous to cytoplasmic kinases. TRPM channels are distributed in various mammalian tissues, predominantly the brain and heart. Notably, however, all TRPM channels except TRPM7 were not expressed in bone and cartilage tissues [77]. The TRPM7 channel has received a lot of attention in recent years due to its essential role in various cellular activities. Studies have shown that TRPM7 plays an integral role in cell proliferation, blood vessel development, and cancer progression [16].

TRPM channel is a widely expressed Ca^2+^ channel, and a potential correlation between TRPM channels and RA FLSs has been demonstrated [78]. Similar to ASIC, the role of the TRPM family subtypes in RA synovial invasion was inconsistent, with TRPM8 being shown to mediate menthol-induced apoptosis in RA rat FLSs [79]. Another member of the TRPM family, TRPM7, is thought to have an inhibitory effect on synovial proliferation. It was noted that TRPM7 can mediate apoptosis in RA FLSs via endoplasmic reticulum stress, and FLSs apoptosis can be inhibited by 2-APB- or Gd^3+^ [80]. Although this difference is acceptable, many experiments are needed to validate 2-APB due to its low selectivity.

During chondrogenesis, the secretion of alkaline phosphatase leads to calcification of the cartilage matrix and chondrocytes apoptosis, causing cartilage traps. Previous studies have shown that silencing TRPM7 reduces collagen expression in cartilage ADTC5 cells and confirms that TRPM7 might be involved in ionic homeostasis during chondrocyte hypertrophy, promoting endochondral ossification [81]. This explains the mechanism of cartilage damage from another perspective, but relevant in vivo experiments are still lacking. Our recent study revealed the role of TRPM7 in cartilage injury in RA. By inhibiting TRPM7 using 2-APB or si-RNA, we demonstrated in vivo and in vitro, respectively, that the inhibition of TRPM7 reduces the extent of cartilage injury in AA rats by modulating Ihh signaling [82]. Interestingly, TRPM7 can not only mediate programmed apoptosis in chondrocytes. Our recent study also revealed that TRPM7 could promote chondrocyte ferroptosis through the regulation of calcium ions, and thus through the PKCα–NOX4 axis. This suggests that the genetic or pharmacological inhibition of TRPM7 could alleviate RA cartilage destruction by reducing chondrocyte ferroptosis [83]. In conclusion, TRPM7 is an effective therapeutic target for RA and the development of more specific drugs will be necessary in future.

Intracellular Ca^2+^ and Mg^2+^ are essential to the proliferation of a wide range of cells. TRPM7, a Ca^2+^ and Mg^2+^ channel, was first shown to be expressed in osteoblasts in 2007 [84]. It was noted that TRPM7 in osteoblasts could be increased in compensation in the absence of extracellular Ca^2+^ and Mg^2+^, and that the specific si-RNA blockade of TRPM7 inhibits osteoblast proliferation [84]. This study reveals an important role for TRPM7 in intracellular ion homeostasis in osteoblasts and suggests that TRPM7 may be an important factor in osteoblast proliferation. Subsequent studies also identified that PDGF mediates osteoblast proliferation and migration via TRPM7-Mg^2+^ influx [85], but interestingly, since intracellular Mg^2+^ plays a regulatory role in TRPM7, interference with TRPM7 in different Mg^2+^ environments could not be excluded. In addition to PDGF, gallium ions can also upregulate osteoblast activity via TRPM7 and promote the expression of osteogenic-related proteins [86]. In recent years, the role of TRPM7 kinase has also received increasing attention. A recent study has clarified the role of TRPM7 kinase in bone regeneration. The research points show that, in the early stages of inflammation, Mg^2+^ can flow into macrophages via TRPM7, leading to the cleavage and nuclear accumulation of TRPM7 kinase and promoting the formation of an immune microenvironment for osteoblasts. In late inflammation, Mg^2+^ promotes the activation of NF-κB in macrophages and upregulates the number of osteoclasts, exerting a negative effect on osteogenesis [87]. Overall, the negative effects of Mg^2+^ on osteogenesis can outweigh its early positive effects. This reveals a diverse role of Mg^2+^ in osteogenesis via TRPM7 kinase. In any case, TRPM7, as a transducer of Mg^2+^ and Ca^2+^, plays an important role in all life activities of osteoblasts and osteoclasts, offering new therapeutic opportunities for diseases related to bone loss.

The signaling pathways regulated by TRPM channels also play an important role in calcium signaling and leukocyte physiology, affecting immune processes such as phagocytosis, degranulation, cytokine expression, chemotaxis and invasiveness. The role that TRPM channels play in the regulation of inflammation in a variety of pro-inflammatory diseases, including RA, has become increasingly clear in recent years [16,88]. Neutrophils, as the most abundant leukocytes, play an important role in immune responses such as inflammation. In RA patients, CD147 has been reported to upregulate neutrophil chemotaxis, adhesiveness and invasiveness, and this role may be due to the influx of Ca^2+^, which is mediated by the activation of TRPM7 [89]. This research reveals the potential role of TRPM7 in the inflammatory response in RA, using promyelocytic leukemia cells and peripheral blood or synovial fluid neutrophils from RA patients with high conviction. In addition, the role of TRPM7 intracellular kinase in the immune response has gained attention in recent years due to the specificity of its structure. Thanks to the development of specific kinase inhibitors, the role of TRPM7 kinase in the inflammatory process of RA has been defined. Studies have shown that neutrophils treated with the TRPM7-specific inhibitor TG100-115 or kinase-activated gene-ablated mouse neutrophils have a significantly reduced ability to respond to Gram-negative bacterial lipopolysaccharide LPS or CXCL8 chemokine gradient migration, and suggest that this effect may be due to the Akt/mTOR pathway [90]. All the above studies suggest that the TRPM7 activity of RA neutrophils is critical to their ability to migrate. In the future, more specific drugs could be developed to target this role and inhibit the inflammatory response in RA.

TRPM7 plays a somewhat deleterious role in RA, and the proper blockade of TRPM7 may slow down the synovitis, as well as the process of cartilage destruction during RA. Unfortunately, however, there is still a lack of clinically relevant drugs that can translate the existing basic research.

### 4.2. TRPC Channel

The TRPC subfamily includes seven members, TRPC1-7. TRPC2 was identified in mice but is a pseudogene in humans [91]. In previous years, the potential role of TRPC in osteoclast genesis has been explored. It was shown that osteoclast formation was increased in I-MFA^−/−^ (MyoD family inhibitor) mice, whereas osteoclast formation was inhibited in mice that were deficient in both TRPC1 and I-MFA genes. Interestingly, they identified a new TRPC1 spliceosome, TRPC1α and TRPC1ε, and revealed that I-MFA might inhibit I_CRAC_ and promote osteoclast formation via TRPC1α and TRPC1ε [92]. In addition to TRPC1, TRPC3 and TRPC6 play important roles in osteoclast regulation. Osteoclast activity is increased in TRPC6-deficient osteoclasts, which exhibit an osteoporotic-like phenotype. In contrast, TRPC3 expression was increased in TRPC6-deficient osteoclasts compared to wild-type (WT), and the inhibition of TRPC3 with Pyr3 reduced bone resorption and differentiation in TRPC6-deficient osteoclasts [93]. This suggests that TRPC6 may play a negative role in osteoclast activity and that the overexpression of TRPC3 promotes bone resorption in osteoclasts. This diverse role has not been independently verified: does reduced osteoclast activity occur in TRPC3^−/−^ mice?

In contrast to TRPM7, TRPC channels were identified as having some anti-inflammatory effects on RA. Studies have shown that mice treated with ML204 (a TRPC5 antagonist) or TRPC5 knockout exhibit greater pain and swlling of joint tissue [94]. However, this study only indicated the anti-inflammatory effect that TRPC5 has on the joints of mice but did not reveal the exact mechanism. More research is needed to validate the role of TRPC channels in the inflammatory response to RA to develop more effective drugs.

### 4.3. TRPV Channel

How do people perceive heat and the stimulation of capsaicin? This question drives the search for tracking temperature receptors under physiological conditions. Finally, in 1997, the founding member of the TRPV family, TRPV1, was discovered to be a heat- and capsaicin-sensitive receptor [95]. In the last 20 years, studies have shown that heat-sensitive TRPVs (TRPV1–4) are widely expressed in various tissues and organs, but are mainly localized to neurons for sensory transmission. Among the TRPVs, TRPV1 is the most representative class of TRPV channels, and its expression in bone tissue has been demonstrated in recent years [96]. In addition, TRPV2 and TRPV4 have been shown to be expressed in synovial tissue [97]. Thermosensitive TRPVs are a class of nonselective ion channels that play important roles in neuropathic pain, inflammation, immunity, neuronal development, diabetes, cardiovascular disease, and cancer by regulating intracellular and extracellular ion homeostasis [98].

#### 4.3.1. TRPV1

TRPV1 was first demonstrated to be significantly more expressed in the FLSs of RA patients than controls in 2007 [99], suggesting an important role for TRPV1 in synovial disorder. Indeed, it was shown that, in a collagen-induced arthritis model, the activation of TRPV1 channels with capsaicin led to the apoptosis of synovial cells through the upregulation of [Ca^2+^]_i_, ROS and mitochondrial membrane depolarization [100].

Mechanical stimulation has long been recognized as a key factor in osteosynthesis, and the role of TRPV channels as a mechanosensitive channels in osteoblasts and osteoclasts has been widely reported. Studies have shown that capsazepine, a TRPV1 ion channel antagonist, inhibits osteoclast bone resorption, osteoblast activity and bone formation, and, in vivo, capsazepine inhibits ovariectomy-induced bone loss in mice [101]. This is the first report on the effect of TRPV1 channels on osteoclasts, showing that the inhibition of TRPV1 has a significantly lower effect on osteoblasts than on osteoclasts. At the overall level, the inhibition of TRPV1 led to a reduction in bone loss, but the exact mechanism behind this has yet to be investigated. This was subsequently demonstrated by Rossi et al., using TRPV1^−/−^ mice [102]. In addition to capsazepine, there are also upstream substances that exert some regulatory effects on osteoclasts via TRPV1. For example, 6-gingerol stimulates TRPV1, promotes the influx of Ca^2+^, and thus stimulates osteoclast differentiation [103]. Sirtuin 1, also known as NAD-dependent deacetylase, has also recently been shown to directly inhibit TNF-α-induced osteoclast genesis in mice by inhibiting TRPV1 [104]. Interestingly, by co-studying the cannabinoid 2 (CB2) receptor on the surface of osteoclasts, it was found that the stimulation of TRPV1 in osteoclasts inhibited osteogenic activity [105,106]. This suggests that the uncertainty in the role of TRPV1 in osteoclasts is due to crosstalk between CB2 receptors and TRPV1. In fact, it has been shown that TRPV1 deficiencies lead to reduced mRNA and protein expression of Runx2 and ALP in bone marrow stromal cells, and reduces osteoclast genesis and osteogenesis [107]. However, this is limited by the fact that the effect of TRPV1 on osteogenic activity has not been studied in osteoblasts. In any case, the important role that TRPV1 plays in osteoclasts has been established, but more research is still needed to investigate the mechanisms behind this and the specific role that TRPV1 plays in osteoblasts in order to provide a therapeutic bullet for RA.

It was noted that knockout of the TRPV1 channel significantly alleviated pain behavior in mice with fully Freund’s adjuvant-induced adjuvant arthritis [108]. Furthermore, a recent drug-controlled trial indicated that the anti-inflammatory effect of using the TRPV1 blocker APHC3 in adjuvant mice could be equal or superior to that of NSAIDs (e.g., diclofenac ibuprofen and meloxicam) [109].

In conclusion, the role of TRPV1 channels in RA is gradually becoming clearer. The activation of TRPV1 may alleviate the synovial invasion process in RA, while the inhibition of TRPV1 may reduce the bone loss process in RA. However, the overall effect on RA is still unclear, and more in vivo experiments will be conducted to verify the therapeutic significance of TRPV1 channel in the regulation of RA.

#### 4.3.2. TRPV2

As a kind of TRP channel, TRPV2 has been shown to be related to the activation of RA FLS in recent years. The data suggest that the specific activation of TRPV2 in FLS can reduce the content of Rac1 in RA FLS and inhibit FLS invasiveness [110]. Further, in vitro experiments have determined that the TRPV2-specific agonist LER13 reduces the invasiveness of FLSs in RA mice, thereby alleviating synovial invasion during RA [97]. These results provide significant evidence for the inhibitory role of TRPV2 in RA synovial membrane invasiveness. In addition to FLSs, there was also a link between TRPV2 and osteoclasts. TRPV2 expression was significantly increased in RAW264.7 cells (preosteoclasts) after RANKL treatment, and the inhibition of TRPV2 significantly reduces the frequency of Ca^2+^ oscillations and osteoclast genesis, revealing an important role for TRPV2 in osteoclast genesis [111]. Later, Bai et al. revealed that TRPV2 regulates osteoclast differentiation through the Ca^2+^-calcineurin-NFAT signaling pathway in multiple myeloma [112]. Further studies in other models, as well as in vivo, are needed to determine the important role that TRPV2 plays in osteoclast differentiation in future. The above studies suggest that TRPV2 may be an essential factor in the differentiation of osteoclasts in RA, and that reducing osteoclast production may alleviate bone loss in RA.

Unlike TRPV1, TRPV2 has been identified as a suppressor of arthritis inflammation. The specific activation of TRPV2 was shown to inhibit the IL-1β-induced expression of MMP-2 and MMP-3 and to suppress the aggressiveness of FLSs [97]. This evidence was also validated by Gulko et al. in FLS cell lines from RA patients [110]. Although the role of TRPV2 in joint inflammation is relatively clear, a lot of research is still needed to explore its specific mechanism in the future.

#### 4.3.3. TRPV4

In addition to TRPV1 and 2, TRPV4 also plays an important role in RA synovitis. It is reported that the activation of TRPV4 following hypotonicity promoted the release of ATP, ROS and cell proliferation from FLSs in RA rats [113]. This suggests that the activation of TRPV4 may aggravate synovial hyperplasia, which is different from TRPV1 and TRPV2, and the reason for this deserves further investigation.

In addition to the synovial invasion, TRPV4 has also been shown to mediate Ca^2+^ influx and chondrocyte apoptosis in osteoarthritis models [114]. In addition to TRPV5, TRPV4 has also been shown to impact the differentiation and activity of osteoclasts [115,116]. It was shown that TRPV4 mediates Ca^2+^ influx into rat osteoclasts and provides sufficient [Ca^2+^]_i_ to maintain RANKL-induced osteoclast differentiation [117]. This study revealed for the first time the important role of TRPV4 in osteoclast genesis and gave rise to the study of TRPV4 in osteoclasts. A recent study also indicated that the knockdown of TRPV4 could inhibit osteoclast differentiation and alleviate osteoporosis in mice by inhibiting cellular autophagy as well as regulating NFATc1 [118]. In any case, the important role of TRPV4 in osteoclast differentiation has been identified and more treatments for osteoclast abnormalities, such as RA or osteoporosis, could be developed, targeting TRPV4 in the future. Furthermore, in vivo double knockout TRPV1 and TRPV4 models also showed reduced osteoclast activity [116], suggesting a role for TRPV1 in osteoclasts. However the researchers did not state whether the effects of TRPV1 and TRPV4 are synergistic or antagonistic. Both genetic and pharmacological inhibition have shown therapeutic potential to alleviate bone loss and structural damage in RA, and more specific drugs targeting TRPV4 will be needed in the future.

TRPV, as a mechanically gated channel, is important in the mechanical nociceptive transmission of the inflammatory process in RA. IL-17, an important inflammatory mediator in the RA process, is thought to upregulate TRPV4 in DRG neurons and mediate nociceptive hyperalgesia [119]. Although the results demonstrate that TRPV4 can play an important role in IL-17-mediated pain, this has not been validated in RA models.

Overall, the activation of TRPV4 may play a role in overall toxicity during RA, including promoting synovitis, mediating osteoclast activation leading to bone loss, and promoting inflammatory bursts. The appropriate inhibition of TRPV4 may alleviate the RA process. However, it is a potential drug target for the treatment of RA.

#### 4.3.4. TRPV5

Ca^2+^ influx is indispensable in chondrocyte apoptosis, necrosis, and matrix degradation. TRPV5, as a Ca^2+^ channel, can mediate Ca^2+^ influx and activate the downstream MAPK and Akt/mTOR pathways, thereby promoting chondrocyte apoptosis [120]. However, the in vivo effect has not been observed using the RA model. Indeed, the role of TRPV channels in mediating cartilage damage in osteoarthritis has been extensively reported [121,122,123], but there is still a lack of RA animal models in which TRPV is the target.

The role of TRPV channels in osteoclast bone resorption was first identified by Eerden BC et al. with TRPV5. It was suggested that TRPV5 channels were expressed in both human and murine osteoblasts and that TRPV5^-/-^ mice showed higher osteoclast numbers and osteoclast volumes compared to WT.; however, bone resorption was virtually absent in TRPV5^−/−^ osteoclast cultures in vitro [124]. This establishes an important role for TRPV5 in osteoclast bone resorption but is limited by the fact that the exact mechanism is not explained. To verify the role of TRPV5 in osteoclast bone resorption, the calcium kinetic regulator econazole was proposed to downregulate TRPV5 levels in rat osteoclasts in a dose-dependent manner and inhibit bone resorption [125]. However, the investigators found an increase in [Ca^2+^]_i_ in osteoclasts after TRPV5 inhibition, suggesting that there may be another intracellular compensatory mechanism that compensates for TRPV5 downregulation, making the results inconclusive. In fact, the important role of TRPV5 in osteoclast bone resorption was questioned in 2008. The data showed that TRPV5^−/−^ mice still underwent significant bone resorption and that treatment with alendronate normalized bone thickness in these mice [126]. Subsequently, it was noted that TRPV5 inactivation completely inhibited the RANKL-induced increase in osteoclast [Ca^2+^]_i_ and was accompanied by a significant activation of bone resorption [127]. This suggests that TRPV5 may be an important pathway, mediating RANKL-induced [Ca^2+^]_i_ elevation in osteoclasts. A role for TRPV5 in osteoclast calcium homeostasis was also confirmed [128]. Interestingly, studies have shown that estrogen inhibits RANKL-induced osteoclast differentiation and bone resorption activity by increasing the expression of TRPV5 channels [129]. Further research revealed that estrogen E2 activates NF-κB via the estrogen receptor, which binds to the 286 nt~−277 nt fragment of the TRPV5 promoter region, upregulates TRPV5 and inhibits osteoclast genesis [130]. The above studies suggest that TRPV5 overexpression may play an inhibitory role in osteoclast genesis and bone resorption, contrary to the results of earlier studies, which may be due to errors caused by objective conditions, such as the use of different animal or cellular models and differences in experimental conditions. In any case, the current research advances suggest that the appropriate stimulation of TRPV5 may reduce bone loss and joint structural damage in late RA.

## 5. P2X7 Receptor (P2X7R)

Adenosine triphosphate (ATP) was originally thought to be the universal energy currency of the cell and has also been identified as an extracellular, non-adrenergic, non-cholinergic (NANC) neurotransmitter in smooth muscle organs [131]. Purinergic signaling is a highly evolutionarily conserved signaling system with extracellular ATP, related nucleotides and adenosine, which serve as transmitter molecules that play a crucial role in many physiological processes and pathological responses [132,133,134]. The receptors stimulated by these purine nucleotides are classified into two types: ligand-gated cation channels, called P2X receptors (P2XRs) (seven mammalian subtypes: P2X1-7), and G protein-coupled P2Y receptors (P2YRs) (eight mammalian subtypes: P2Y1, 2, 4, 6, 11-14) [135,136]. P2XR mediates the inward flow of extracellular cations, including Na^+^, Ca^2+^ and K^+^, after binding to ligands.

It has been shown that P2X7R plays a potential role in the regulation of RA development and progression [137,138]. At the genetic level, studies have shown that polymorphisms at positions 1068 and 1513 of the P2X7R gene may be susceptibility loci for RA [139]. Comparable research has demonstrated that the 489C>T SNP in the P2X7R gene potentiates the function of P2X7R in RA patients and may be engaged in the development of RA [140]. In addition, the significantly elevated expression of P2X7R in the peripheral blood mononuclear cells of RA patients has been observed in several studies [141,142]. In other cases, elevated P2XR levels in synovial tissue have been found in patients with RA, which has a high diagnostic value for RA [133,143]. The above studies suggest that P2X7R plays an important role in the pathogenesis of RA. In cartilage injury, it has been suggested that the over-activation of P2X7R can mediate apoptosis in osteoarthritic articular chondrocytes through the IRE1-mTOR-PERK axis [144], but can this result be extended to RA? A significant number of in vitro and in vivo studies are needed to demonstrate the specific mechanisms of P2XR in articular cartilage damage in RA.

ATP was shown earlier to be a potent stimulator of osteoclast formation, and P2XR, a purinergic signaling receptor, and was also found to play an important role in osteoclasts in recent years [145]. Osteoblasts and osteoclasts express a range of P2X purinergic receptors. However, until recently, only the P2X7R subtype’s role in the bone metabolism has been documented, and fewer studies have addressed the role of the remaining six P2X receptor subtypes [146]. The role of P2X7R in osteoblasts is still controversial. To mimic osteoporosis after estrogen withdrawal, the use of ovariectomized mice revealed more severe bone loss in P2X7R-deficient mice, and histological examination showed an increased number of osteoclasts in P2X7R^−/−^ mice [147]. This is the first in vivo demonstration of the activating effect of inhibiting P2X7R on osteoclasts and is consistent with previous studies at the genetic level [148,149]. In addition, appropriate concentrations of ATP were shown to activate P2X7R and mediate Ca^2+^ inward flow in rabbit osteoclasts, possibly leading to the inhibition of bone resorption [150]. The above studies suggest that P2X7R activation exerts a negative regulation on bone resorption. Surprisingly, blocking P2X7R on osteoclast precursors with blocking antibodies inhibited the formation of multinucleated osteoclasts in vitro, while in vivo P2X7R knockout mice maintained the ability to form multinucleated osteoclasts [151]. This discrepancy may be due to differences in experimental models or to the complexity of the system. Further studies have shown that the P2X7R antagonists AZ15d, KN62 and OATP all reduce RANKL and M-CSF-induced osteoclast genesis in vitro [152]. However, it is worth noting that the P2X7R antagonists may activate other mechanisms, such as AZ15d, which can affect some pore structures, activated by trichothecene [153]. A recent report also showed that surface expression of P2X7Rs was downregulated when RAW 264.7 cells were chronically exposed to high levels of ATP, thereby preventing precursor osteoclast fusion [154]. In conclusion, the present findings demonstrate that either the activation or inhibition of P2X7R can have an inhibitory effect on osteoclast genesis, which may be the result of the combined action of multiple P2XR isoforms due to the low experimental selectivity. In fact, previous studies suggest that low concentrations of ATP can activate P2XRs other than P2X7 receptors, such as P2X2R and P2X3R, thereby increasing bone resorption [145]. It was shown that the selective non-nucleotide antagonist of the P2X2/3 receptor, A317491, reduced RANKL expression in mice in an unloading model with tail suspension [155], and, in addition, 1 µM Minodronic acid (an antagonist of P2X2R and P2X3R) treatment of mice resulted in t hedetachment of osteoclasts from the bone surface, inhibiting bone resorption [156]. These studies reveal a potential role for P2X2/3R in osteoclast formation and bone resorption, and more research is needed in the future to confirm the mechanisms behind this.

As a purinergic receptor, P2XR is involved in a variety of autoimmune diseases and plays an important role in the regulation of immune processes. In the regulation of inflammation by P2XR, it has been suggested that P2X7R enhances the release of pro-inflammatory factors by macrophages [157]. In addition to macrophages, IL-17-producing T helper (Th17) cells are the main effector cells in the pathogenesis of RA, and recent studies have shown that the pharmacological inhibition of P2X7R in a model of collagen-induced experimental arthritis exhibits a significant reduction in Th17-cell-promoting factors (e.g., IL-1β and IL-6) [158]. This clarifies that P2X7R plays an important role in the release of inflammatory factors in collagen-induced arthritis and in T cell differentiation, and suggests a new approach to P2X7R-mediated inflammation. In fact, in early years, researchers suggested that there may be a cytokine-independent pro-inflammatory mechanism of the P2X7R that mediates the release of histone proteases from macrophages and suggested that this mechanism may be relevant to joint disease [159]. Although the role of P2X7R in the regulation of peripheral inflammation in RA is relatively well-established, its specific mechanisms need to be further explored and may involve multiple cellular interactions.

Overall, P2X7R plays an important role in the structural damage of RA, and plays a certain toxic role in cartilage apoptosis, suggesting that the inhibition of P2X7R can alleviate RA to a certain extent. However, relevant clinical research is still lacking at present, and some translational research could be appropriately carried out in the future (Figure 3).

## 6. Other Channels

### 6.1. Orai1 Channel

Upon depletion of intracellular Ca^2+^, store-operated calcium (SOC) channels mediate inward flux of extracellular Ca^2+^ to replenish the Ca^2+^ pools of the endoplasmic reticulum. This phenomenon was proposed more than 30 years ago by Putney et al. [160]. The most calcium-selective of the SOC channel family is the Ca^2+^ release-activated-Ca^2+^ (CRAC) channel, which mainly consists of Orai channels in the cytoplasmic membrane and stromal interaction molecules (STIM) in the endoplasmic reticulum membrane [161]. There are three mammalian members of the Orai channel (Orai1-3), which mainly consists of a hexametric complex forming a selective calcium-permeable channel in the cytosol, of which Orai1 is the most intensively studied, and a 33 kDa protein consisting of four transmembrane regions, with both the N and C termini located intracellularly [162]. Orai1 channels activated by STIM1 play an important role in a variety of pathological and physiological processes. In the central nervous system, mice knocked out of Orai1 or its activator STIM1 exhibited stronger seizures, as well as convulsive mortality [163]. Orai1-STIM1 also plays an important role in tumors. For instance, the downregulation of Orai1 reduces the invasion of glioblastoma cells and downregulates the proliferation of breast cancer cell lines [164].

In 2014, the Orai1 channel was shown to be genetically polymorphic in the Orai1 gene and associated with RA susceptibility in the population, linking Orai1 etiologically to RA [165]. Furthermore, the efficacy of using si-RNA intra-articular injections in RA has been established [166,167]. Although Orai1 has been shown to be associated with susceptibility to RA, information on its role in RA synovial invasion remains relatively limited. A recent report suggests that Orai1 channels can reverse Fas apoptotic signaling in RA FLSs [168]. It has been suggested that Orai channels may play a synovial protective role in RA, and further exploration of the role that CRAC channels play in RA synovial invasion is needed in the future to develop more promising therapeutic targets. In addition to its effect on synovial inflammation, the role that Orai1 channels play in osteoclasts is becoming increasingly evident, suggesting its place in RA joint damage [169]. In the previous decade, it was shown that Ca^2+^ oscillations in pro-osteoclasts were triggered by the RANKL-dependent activation of TRPV2 and SOCE and intracellular Ca^2+^ release [111]. Furthermore, STIM1 and TRPV4 have also been shown to regulate shear-induced calcium oscillations during osteoblast differentiation [170], suggesting a potential role for Orai1 in intracellular calcium homeostasis in osteoclasts. Later, by observing shRNA silencing of Orai1 in RAW264.7 cells, NFATc1 transduction was found to be inhibited, with impaired osteoclast synthesis [171]. This is direct evidence of an important role for Orai1 in osteoclasts and suggests that this may be due to the silencing of Orai1 leading to the inhibition of SOCE, and consequently affecting the transcription of RANKL signaling. Interestingly, another study found reduced cortical ossification and trabecular thinning in Orai1^−/−^ animals compared to controls on microcomputed tomography, and subsequent in vitro studies confirmed that the inhibition of Orai1 activity not only impairs the differentiation of human osteoclasts, but of osteoblasts and their function [172]. It is hypothesized that the loss of Orai1 in osteoblasts leads to calcium disorders; therefore, silencing Orai1 can have an effect on both osteoblasts and osteoclasts. Could this mechanism be extrapolated to chondrocytes? More research will be needed in the future to confirm the role of Orai1 in chondrocytes.

Overall, research on Orai1 in RA is lacking at present, although it has shown some results in terms of RA synovial protection and bone loss (Figure 4).

### 6.2. Piezo1 Channel

In mammals, mechanical signal transduction mainly involves physiological responses to touch, pressure, airflow, sound, blood pressure, etc. Mechanical transduction often requires mechano-transduction proteins to convert mechanical signals into biochemical signals. The discovery of a new group of mammalians, mechanically activated channels, Piezo channels, by Patapoutian et al. in 2010 attracted considerable attention from researchers around the world and opened the field of research into eukaryotic mechanically activated channels [173]. There are two Piezo proteins (Piezo1–2) in mammals and, along with their massive structures, they are the largest cytosolic ion channel complexes reported to date [174]. Human Piezo1 channels are permeable to monovalent ions such as K^+^, Na^+^ and Cs^+^, divalent ions such as Ba^2+^, Ca^2+^ and Mn^2+^, and organic cations such as tetramethylammonium and tetraethylammonium.

The potential role that Piezo channels play in chondrocytes was first proposed by Lee et al. [175]. Subsequently, the Piezo1 protein was identified as having an apoptosis-inducing effect on human primary chondrocytes, possibly via the MAPK/ERK signaling pathway [176]. This study clarifies the mechanism by which mechanical loading leads to chondrocyte apoptosis and lays the foundation for explorations of the role of Piezo1 channels in degenerative cartilage diseases. Although the mechanism by which the Piezo1 channel mediates chondrocyte apoptosis has been well-established in models of osteoarthritis [177], studies in RA animal models have been very limited. In addition to its effect on chondrocyte apoptosis, a recent study found that targeting Piezo1 deficiency in chondrocytes resulted in an osteoporotic phenotype and further revealed that Piezo1 deficiencies impaired the process of chondrocyte ossification, leading to impaired bone trabecula synthesis [178]. This suggests that although Piezo1 mediates chondrocyte apoptosis, its in vivo deletion can also lead to impaired chondrocyte ossification and consequently increased bone loss. This paradoxical effect has led to difficulties in drug development and more in vivo experiments are needed to investigate the combined effects of Piezo1 on articular cartilage in vivo to better develop therapeutic agents for RA.

Piezo1 is expressed as a mechanosensitive channel in a variety of non-sensory tissues, and mechanical forces are also thought to be an effective stimulus for bone reconstruction; therefore, is there a role for Piezo1 in osteosynthesis? A genetic impact mapping of human osteoporosis identified several Piezo1 single-nucleotide polymorphisms associated with osteoporosis and fracture [179], suggesting a potential role for Piezo1 in osteosynthesis. In addition, in human primary periodontal ligament cells (hPDLC) cultured in vitro, Piezo1 expression increased after mechanical stress stimulation and persisted for 12h, and osteoclast-activation-related genes, such as RANKL and COX2, were activated, whereas RANKL and COX2 were inhibited after the use of a Piezo1 inhibitor (GsMTx4). To verify the effect of Piezo1 modulation on osteoclasts in hPDLC, the co-culture of hPDLC and RAW264.7 showed that the RAW264.7’s ability to differentiate into osteoclasts was reduced after GsMTx4 treatment [180]. The results of this study suggest another important role for Piezo1 in osteoclast synthesis, but the current studies are limited to in vitro studies, and the exact mechanism is not yet clear. It is worth noting that a recent in vivo study reported increased bone resorption in Piezo1 conditional knockout mice, mechanistically suggesting that Piezo1 deficiency impairs Col-II and Col-IX production by reducing YAP nuclear localization, leading to increased osteoclast activity [181]. This seems to be the opposite conclusion to the previous in vitro experiments, but the in vivo experiments are more convincing as there are many reasons for the in vitro errors, such as a lack of high selectivity in the GsMTx4, failure to consider the complexity of the internal environment, and the fact that Jin et al. did not directly demonstrate the role of Piezo1 in osteoclasts. Further research is needed to explore the other effects of Piezo1 on osteoclasts and its mechanisms to better treat osteoclast abnormalities in diseases such as RA (Table 1).

## 7. Concluding Remarks

In this review, we briefly describe the main pathological features of RA, including cartilage damage, synovial invasion and the onset of joint inflammation, and present a short description of the calcium-permeable channels, such as ASIC, TRP, P2XR and Piezo, which are involved in the pathogenesis of RA. Finally, we explain the role of these calcium-permeable channels in the pathogenesis of RA. At the overall level, the activation of the vast majority of Ca^2+^-permeable channels can exacerbate RA synovial inflammation by promoting synovial cell hyperproliferation, as well as other signaling pathways. However, a small number of channels can also promote synovial cell apoptosis. In addition, at the level of structural damage in RA, the activation of Ca^2+^-permeable channels often leads to chondrocyte death and osteoclast activation through different signaling pathways. This subsequently leads to increased cartilage damage and bone erosion in RA. Although the specific mechanisms of some of these channels in the pathogenesis of RA are still unclear, there are sufficient studies supporting their role in RA. The description of the role of calcium-permeable channels in various aspects of RA provides a more comprehensive and in-depth review of the therapeutic targets in RA.

## Figures and Tables

**Figure 1 biomolecules-12-01383-f001:**
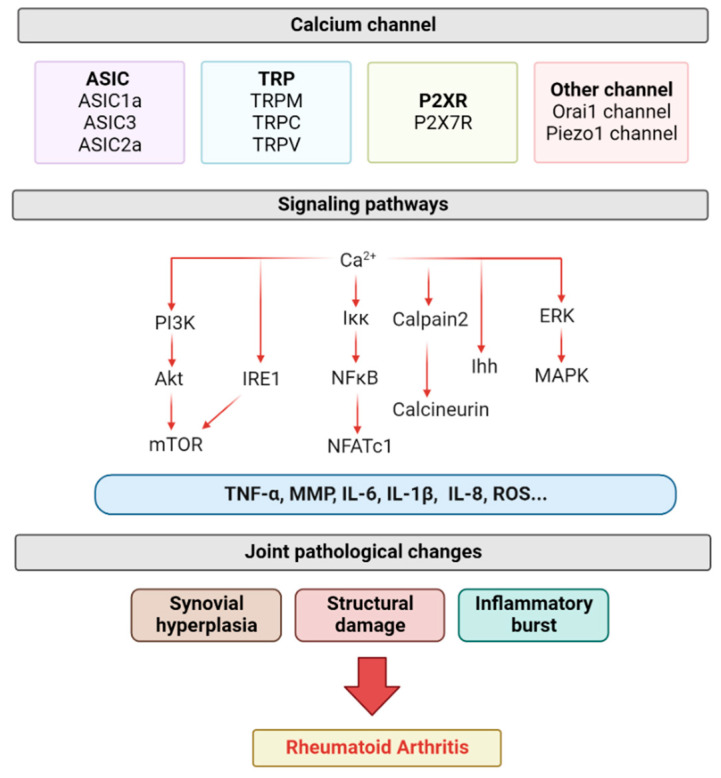
The role of calcium-permeable channels in the pathogenesis of RA. ASIC, TRP, P2X7R, Orai1 and Piezo1 channels regulate a variety of calcium-ion-related signaling pathways, which, in turn, lead to changes in related inflammatory factors, ultimately leading to synovitis, structural damage, and inflammatory outbreaks (created with Biorender.com, access date 29 July 2022).

**Figure 2 biomolecules-12-01383-f002:**
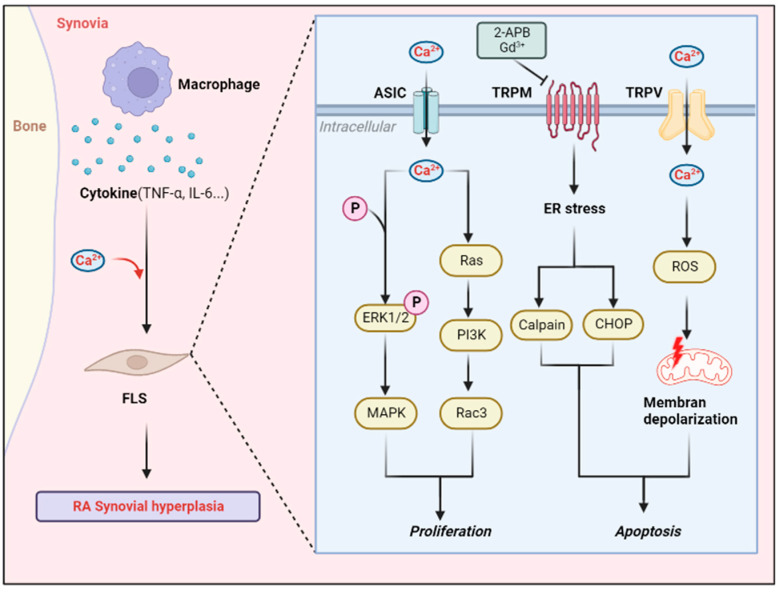
Role of calcium-permeable channel in synovitis. Macrophages can secrete a variety of cytokines to stimulate FLSs in the synovium, leading to a large influx of calcium ions in FLSs cells and triggering related cascade reactions, eventually leading to excessive apoptosis or proliferation of FLSs, and regulating synovial inflammation (created with Biorender.com).

**Figure 3 biomolecules-12-01383-f003:**
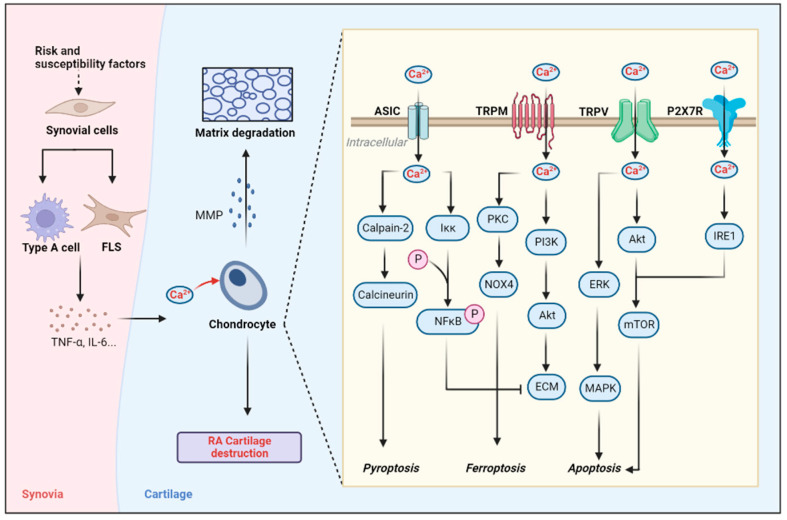
Role of calcium-permeable channel in cartilage destruction. Stimulated synovial cells can promote chondrocyte apoptosis by secreting inflammatory factors. At the same time, a large influx of calcium ions in chondrocytes triggers the activation of intracellular calcium ion signaling pathways (NF-κB pathway, PI3K-Akt pathway, etc.). This eventually leads to the degradation of cartilage matrix and apoptosis, pyroptosis and ferroptosis of chondrocytes (created with Biorender.com).

**Figure 4 biomolecules-12-01383-f004:**
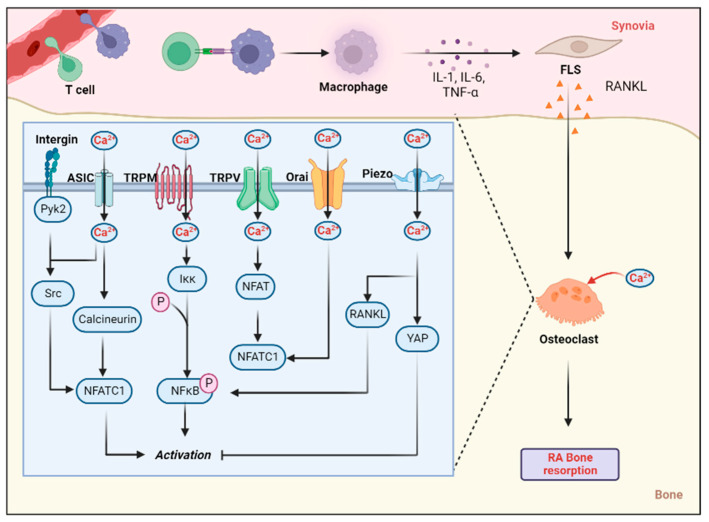
Role of calcium-permeable channel in bone resorption. Synovitis-activated FLS cells can release a number of factors, such as RANKL, leading to an influx of osteoclast calcium ions in the bone. Through signal transmission, this eventually leads to the proliferation and activation of osteoclasts, which intensifies bone resorption (created with Biorender.com).

**Table 1 biomolecules-12-01383-t001:** Role of activating the calcium-permeable channel in RA.

		Synovial Hyperplasia	Cartilage Destruction	Bone Damage	Inflammation
ASIC	ASIC1a	RA synovial invasion, Ras-associated C3 botulinum toxin substrate1 [48]; RA FLSs proliferation, ERK/MAPK [49,50]	ECM expression [53]; chondrocyte pyroptosis, calpain-2/calcineurin [56,57]; chondrocyte apoptosis, NF-κB [23]	osteoclast migration, integrin/Pyk2/Src [61]; osteoclast genesis, NFATc1 [63]	MIP-1a, IL-8, NFATc3/RANTES [49]
	ASIC2a		chondroprotective effects of PcTx1 and APETx2, ERK/MAPK [64]		
	ASIC3	FLS apoptosis, [Ca^2+^]_i_ [55]; Sensitivity of FLS to inflammatory mediators [69]	hyaluronic acid [68]		IL-6, MMP [69]; Pain behavior [70]
TRPM	TRPM7	FLS apoptosis [80]	Ihh, chondrocyte damage [82]; collagen X, PI3K/Akt [81]; ferroptosis, PKCα-NOX4 [83]	osteoclast genesis [85]; osteoblast proliferation [84]	neutrophil chemotaxis, adhesion, invasiveness, Ca^2+^ influx [89]; Akt/mTOR [90]
	TRPM8	FLS apoptosis [79]			
TRPC	TRPC1			osteoclast genesis [92]	
	TRPC5				joint pain and swelling [94]
	TRPC6			osteoclast activity [93]	
TRPV	TRPV1	FLS apoptosis, [Ca^2+^]_i_, ROS mitochondrial membrane depolarization [100]		osteoblast differentiation [103]; bone resorption [101]	painful behavior [109]
	TRPV2	aggressiveness of FLS [97]		osteoclast differentiation, Ca^2+^-NFAT signaling [111]	MMP-2, MMP-3 [97]
	TRPV4	proliferation of FLSs [113]	chondrocyte apoptosis [114]	osteoclast differentiation, RANKL [117]; NFATc1 transcription, cellular autophagy [118]	
	TRPV5		chondrocyte apoptosis, MAPK and Akt/mTOR pathways [120]	osteoclast genesis [130]; osteoclast [Ca^2+^]_i_, bone resorption [129]	
P2XR	P2X7R		chondrocyte apoptosis, IRE1-mTOR-PERK [144]	osteoclasts activation [150]; number of osteoclasts [147]	histone proteases [159]; IL-1β, IL-6 [158]
Other channel	Orai1	Fas signaling [168]		osteoclast synthesis, NFATc1 transduction [171]; osteoblast function, bone trabeculae [172]	
	Piezo1		chondrocyte apoptosis, MAPK/ERK [177]	RANKL, RUNX2, osteoclasts differentiation [180]; osteoclast activity, bone resorption [181]	

i: intracellular.

## Data Availability

Not applicable.

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
