# Peer review of "Calcium-Permeable Channels Cooperation for Rheumatoid Arthritis: Therapeutic Opportunities"

_biomolecules, 2022, doi:10.3390/biom12101383_

Round 1

Reviewer 1 Report

This is a short, but still comprehensive overview about calcium channels and their function in RA. Although it is listing studies of calcium channels and their function in RA, the overall connection of these channels to RA could be more detailed and better explained and connected. Furthermore, the introduction part should explain the importance and role of calcium ions themself in RA, also making the connection to the contribution of calcium in inflammation, especially also in RA. The review focuses mainly on osteoclasts in RA, this cell type should therefore also be introduced briefly. The graphical summaries are very helpful for getting an overview, but more details would help to better understand the contribution of each channel in RA and which signaling pathways lead to e.g. osteoclast activation/proliferation or apoptosis, these connections should be added. Therapeutic opportunities should be generally more highlighted. 

The authors should generally avoid sentences longer than 2-3 lines and the description of different studies in one sentence. Several sentences need to be changed/proved, e.g. line 72-75, 156/157, end of 199, 306/307, 324, 376-378, 419-421, 632-636. 

Reviewer 2 Report

Hello.

The proposed scheme for presenting the material is classical and reflects the adverse events that occur against the background of the use of known drugs for the treatment of RA.

In the future, the material combines and reveals new approaches to the search for targets based on permeable ion channels.

The main novelty of the work is the integration of disparate data on physiology and pharmacology.

The proposed review will be useful for understanding the role of ion channels in the formation and pathogenesis of RA with a possible subsequent correction of the condition.

An overview of the three main groups of ASICs, TRPV and P2X provides insight into the concept and role of each in pathogenesis.

Additionally, channels are reflected that are not sufficiently studied and are new undeciphered targets.

I believe that the review is applicable to the special issue and can be published.

Remarks, do not match the source of literature and references in Table 1. Please double-check.

Reviewer 3 Report

The article "Calcium Permeable Channels Cooperation for Rheumatoid Arthritis: Therapeutic Opportunities" is a review about the importance of calcium channels in relation to rheumatoid arthritis. It has a lot of inaccuracies about rheumatoid arthritis; its pathogenesis, clinic and current treatment. Research on the dynamics of calcium channels is important in autoimmune diseases, however, the approach is not correct. An extensive article that does not adequately guide the reader about what is known about the subject and its particular importance in rheumatoid arthritis. Many pathophysiological aspects are not properly oriented.

In conclusion, it is not a suitable article for publication.

Round 2

Reviewer 3 Report

Many pathophysiological aspects are not properly oriented.. Requires orientation regarding pathophysiology of RA. Say for example that "In RA, the deposition of autoantibodies causes the synovium to change from a normal phagocytic structure to a tissue with a large infiltration of inflammatory cells (including T cells, macrophages, B cells, etc.)" is not correct. Or to say that patients currently do not have adequate medication for their treatment and that adverse reactions are a major problem, is not correct either.
